# Characterization and Molecular Dynamics Simulation of a Lipase Capable of Improving the Functional Characteristics of an Egg-Yolk-Contaminated Liquid Egg White

**DOI:** 10.3390/foods12224098

**Published:** 2023-11-11

**Authors:** Linlin Xu, Fei Pan, Yingnan Li, Huiqian Liu, Chengtao Wang

**Affiliations:** 1Beijing Advanced Innovation Center for Food Nutrition and Human Health, Beijing Engineering and Technology Research Center of Food Additives, Beijing Technology & Business University (BTBU), Beijing 100048, China; xll18515351711@163.com (L.X.); q2223599235@163.com (H.L.); 2State Key Laboratory of Resource Insects, Institute of Apicultural Research, Chinese Academy of Agricultural Sciences, Beijing 100093, China; yunitcon@yeah.net; 3Ministry of Education Key Laboratory of Industrial Biotechnology, School of Biotechnology, Jiangnan University, Wuxi 214122, China; yingnan3400189@gmail.com

**Keywords:** lipase, foaming property, molecular dynamics simulation, thermal stability, dynamic cross-correlation analysis

## Abstract

Lipase has great application potential in hydrolyzing residual yolk lipid in egg white liquid to restore its functional properties. In this study, a lipase gene from *Bacillus subtilis* was expressed in *E. coli* BL21 (DE3) and named Lip-IM. Results showed that although Lip-IM has stronger specificity for medium- and short-chain substrates than long-chain substrates (C16, C18), due to its excellent enzyme activity, it also has strong hydrolysis activity for long-chain substrates and maintained over 80% activity at 4–20 °C, but significantly reduced when the temperature exceeds 40 °C. The addition of 0.5% Lip-IM enhanced foaming ability by 26% (from 475 to 501%) and reduced liquid precipitation rate by 9% (from 57 to 48%). Furthermore, molecular dynamics (MD) simulations were run to investigate the conformational stability of Lip-IM at different temperatures. Results showed that Lip-IM maintained a stable conformation within the temperature range of 277–303 K. Fluctuations in the flexible area and backbone movement of proteins were identified as the main reasons for its poor thermal stability.

## 1. Introduction

Liquid egg products include whole egg liquid, egg white liquid, and egg yolk liquid. Among them, egg white liquid is widely used in food production due to its high foaming performance, which is unmatched by other animal and plant proteins [1]. Egg white liquid is essentially an aqueous protein solution that, when whipped, can increase in volume by 6–8 times and improve the texture properties of baking batter [2]. The great foaming properties of egg white liquid make it an indispensable ingredient in baked goods such as puffy omelets, meringues, and angel cakes [3]. However, the low concentration of yolk residue in egg white solution negatively affects the foaming characteristics of egg white liquid, significantly impacting the quality of baked goods [4]. Wang and Wang [5] reported that lower concentrations of triacylglycerol (0.022%) reduce foaming ability. In addition, the presence of egg yolk residue exerts a more significant impact on foaming performance compared to pasteurization and storage. Nielsen [6] found that the residual amount of egg yolk ranged from 0.005 to 0.4% in commercial egg-separation operations. Furthermore, during egg beating and splitting operations, egg yolk residues in egg white liquid can arise due to poor egg quality, easy breakage of the eggshell, and erroneous operation. Current separation techniques fail to ensure full isolation of yolk and egg white, leading to negative impacts on the foaming performance of egg white, and hence, limiting the progress of the liquid egg industry. Therefore, it is imperative to develop effective separation techniques or strategies that can minimize the residual yolk components in egg white liquid.

At present, various approaches, including physical, chemical, and enzymatic methods, are employed to enhance the foaming characteristics of egg white. Among these, the structure of proteins has been extensively modified by physical techniques, such as heat treatment, ultraviolet irradiation, and dynamic high-pressure treatment [7,8]. Meanwhile, chemical techniques, such as Maillard glycation, can enhance the surface hydrophobicity and adsorption speed of the protein interface to improve the foaming ability [9]. Enzymatic methods, including the use of neutral protease, alkaline protease, papain, and trypsin, involve enzymatic reactions that enhance the protein structure or molecular weight. They improve the foaming characteristics of egg white by changing its surface hydrophobicity and adsorption speed [10]. However, studies have found that trypsin treatment may not necessarily improve foam stability. According to Zhao et al. [11], an increase in protease concentration increased foam particle size, accompanied by a more uneven particle size distribution. This suggests that although the protease hydrolysate improved the foaming ability to some extent, the foam stability was not enhanced. When each group of egg white liquid was made into cakes, products containing 5% protease hydrolysate collapsed due to uneven bubble diffusion. To effectively remove lipid components from residual egg yolk contamination, centrifugation, lipase, and phospholipase methods have been utilized. Cunningham and Cotterill [12] reported that residual egg yolk lipids in egg white liquid can be removed by centrifugation to restore their functional properties. Lipase and phospholipase, on the other hand, can remove lipid components from the residual egg yolk by hydrolyzing triglycerides and phospholipids, respectively. Kobayashi et al. [13] have successfully demonstrated the feasibility of utilizing immobilized lipases to enhance the foaming performance of egg white liquid with residual yolk. After a treatment of 30 min with two immobilized lipases, the foaming stability of egg white solutions with residual 0.05% yolk was restored.

Lipases (EC 3.1.1.3) are a unique class of enzymes capable of hydrolyzing long-chain triglycerides into fatty acids, diacylglycerol, and glycerol [14]. Moreover, lipases can catalyze a variety of reactions such as esterification, alcoholysis, transesterification, and acidolysis in the non-aqueous phase. Lipases are ubiquitous in animals, plants, and microorganisms, with microbial lipases being more diverse, displaying excellent enzymatic properties and ease of large-scale production when compared with animal and plant lipases. Therefore, microbial lipases are an essential industrial source that have extensive applications in many fields, such as the food industry, chemical industry, and medicine, due to their catalytic diversity and substrate specificity [15].

Bacillus subtilis lipase A (BSLA) is a small molecular weight lipase with a simple structure that exhibits excellent potential for industrial applications and has been used as an ideal model for protein calculation and design. In this work, we expressed the Bacillus subtilis lipase gene in E. coli. BL21 (DE3), designated Lip-IM, and purified the enzyme to perform biochemical property analysis and kinetic parameter determination. We discovered that although Lip-IM exhibits stronger specificity towards medium- and short-chain substrates than long-chain substrates, it possesses exceptionally high specific activity. Consequently, its activity towards long-chain substrates is superior to that of most other lipases, which is advantageous in the presence of egg yolk lipids containing a significant amount of 16:0, 18:0, and 18:1n-9, accounting for over 75% of the total fatty acid content in the yolk [16]. Furthermore, Lip-IM can effectively restore the foaming property of egg white liquid, making it a viable option for application within the liquid egg industry. To improve the efficiency of enzymatic hydrolysis of lipases, biological modifications were considered, and molecular dynamics (MD) simulations were conducted to investigate the conformational stability of Lip-IM at different temperatures. This study aimed to explain protein stability and residue interaction at different temperatures, thereby deepening the understanding of the relationship between temperature and the structure–function of lipases. The findings of this study provide theoretical guidance for subsequent biological modifications of Lip-IM to obtain highly efficient lipases with high catalytic activity, offering a new perspective in the development of new enzyme preparations.

## 2. Materials and Methods

### 2.1. Materials

The pET-28a (+) vector and *E. coli* BL21(DE3) expression host were obtained from General Biol (Chuzhou, China). Restriction endonucleases (Hind III, Bam HI, T4 DNA ligase) were bought from TaKaRa. Isopropyl β-D-Thiogalactoside (IPTG), p-nitrophenyl caproate (pNPC, C6), p-nitrophenyl caprylate (pNPC, C8), p-nitrophenyl decanoate (pNPD, C10), p-nitrophenyl laurate (pNPL, C12), p-nitrophenyl myristate (pNPM, C14), p-nitrophenyl palmitate (pNPP, C16), and p-nitrophenyl stearate (pNPS, C18) were bought from Aladdin Industrial Corporation. Fresh eggs were bought from Beijing Deqingyuan Agricultural Technology Co., Ltd. (Beijing, China).

### 2.2. Gene Expression and Protein Purification

The coding sequence of Lip-IM was obtained from NCBI (FJ544454.1). The recombinant Lip-IM expression vectors were constructed by cloning the Lip-IM coding sequence into pET-28a (+) using Hind III and Bam HI restriction endonucleases. The recombinant vector was then transformed into an *E. coli* BL21(DE3) strain.

Recombinant bacteria were inoculated into 3mL LB with 3uL kanamycin, and then incubated at 37 °C for 16 h. Then the culture was inoculated into 50 mL fresh LB with 50 uL kanamycin in the proportion of 2%. When OD600 of the culture reached 0.6–0.8, Isopropyl β-D-Thiogalactoside IPTG with a final concentration of 0.4mM was added to induce lipase expression. After incubation for 16 h (20 °C, 200 rpm), the culture was centrifuged (10,000 rpm, 4 °C, 5 min) and the bacteria collected; the supernatant was discarded and the bacteria were resuspended with an appropriate amount of lysis buffer. The resuspended bacteria were broken by ultrasonic cell fragmentation apparatus on an ice bath, and the supernatant and precipitation were separated by centrifuging (10,000 rpm, 4 °C, 5 min), with the supernatant being the crude enzyme solution.

The 6×His-tagged recombinant lipase was purified using Ni-NTA beads 6FF. The crude enzyme solution was filtered using 0.22-μm filter membranes, the miscellaneous protein was washed by washing buffer, and the target protein was separated from the miscellaneous protein using elution buffer with different imidazole concentrations. Finally, the target protein was collected. The imidazole present in the purified protein was eliminated by dialysis in 50 mM NaH2PO4 (pH = 8). The Bradford method was used to determine the protein concentration. The purified protein was analyzed using 12% SDS-PAGE. Protein concentration was quantified using the Bradford Protein Concentration Kit.

### 2.3. Enzyme Assay

#### 2.3.1. Enzyme Activity Assay

The steps of p-NPP method are as follows: 0.2 mL of substrate solution (the final concentration of p-NPP in isopropanol is 10 mM) was added to 1.5 mL buffer (50 mM Tris HCl, pH 8) in tubes, and preheated at 37 °C for 15 min, and then added 0.1 mL of appropriately diluted enzyme solution, mixed and reacted at 37 °C for 15 min, the absorbance value of the reaction mixture at 410 was measured after adding 1 mL of SDS to terminating the reaction, and each experiment was made in triplicate. The inactivated enzyme was used as control.

Enzyme activity definition: one unit (U) of enzyme activity was defined as the amount of enzyme required to catalyze p-NPP to release 1 μm p-NP per minute under certain conditions.

Enzyme specific activity definition: indicates the purity of the enzyme and represents the number of enzyme activity units per milligram of protein (U/mg).

#### 2.3.2. Effect of Temperature on Enzymatic Properties

To estimate the optimum temperature of Lip-IM, the enzyme solution was appropriately diluted and mixed with the substrate as per the aforementioned method. The mixture was incubated at 4, 20, 30, 40, 50, and 60 °C for 15 min in buffer (50 mM Tris-HCl, pH 8). Enzyme activity was measured at each temperature. To assess the stability of Lip-IM, the enzyme was incubated at the above temperatures for 3 h. At 30-min intervals, the enzyme was mixed with the substrate, under standard conditions, and the remaining enzyme activity was measured [17].

#### 2.3.3. Effect of pH on Enzymatic Properties

The enzyme activity was measured in different pH buffers under the optimal temperature conditions mentioned above, to determine the optimal pH of enzymes [18]. The buffer solutions were configured as follows: citric-acid–phosphate buffer (pH = 6–7), Tris-HCl buffer (pH = 8–9), and sodium-carbonate–sodium-bicarbonate buffer (pH = 10). The enzyme solution was incubated in the above buffers for 12 h, respectively, and the pH stability of the lipase was assessed by determining the residual activity at 30 °C.

#### 2.3.4. Effect of Metal Ions on Enzymatic Properties

To investigate the impact of metal ions on Lip-IM activity, different metal ions were added to the lipase solution. The metal ion concentrations were varied as 1mM and 10mM, including Na^+^, K^+^, Ca^2+^, Cu^2+^, Mn^2+^, Zn^2+^, Mg^2+^, Fe^2+^, Cr^3+^, and Fe^3+^. Under optimal conditions, the residual lipase activity was determined after 1 h of incubation, with the control sample being the enzyme without any added metal ions [19].

#### 2.3.5. Substrate Specificity

To assess Lip-IM activity on different substrates, the following p-nitrophenyl esters were used: caproate (C6), caprylate (C8), decanoate (C10), laurate (C12), myristate (C14), palmitate (C16), and stearate (C18). All experiments were carried out under the optimum pH and temperature.

#### 2.3.6. Effect of Chemical Solvents on Enzymatic Properties

In this experiment, methanol, ethanol, isopropanol, acetonitrile, dimethyl sulfoxide (DMSO), SDS, EDTA, Tween-80, and Triton X-100 were added to the reaction system with final concentrations of 5% and 10%, respectively. Under optimal temperature and pH conditions, the residual lipase activity was determined after 1 h of incubation, and the enzyme without chemical solvent was used as the control [20].

#### 2.3.7. Determination of Enzymatic Kinetic Parameters

Prepare the substrate with a concentration of 0.01–30 mM and determine the lipase activity under the optimum conditions. According to the above method, the change in absorbance value was measured at OD410, and each experiment was made in triplicate [21]. Km and Vmax were calculated through nonlinear regression of the Michaelis–Menten equation using GraphPad Prism 8.0.

### 2.4. Effect of Lip-IM on Foaming Properties of Egg White System

#### 2.4.1. Sample Prepared

The yolk was separated from the egg white carefully. Five different EW–EY solutions were prepared: the egg white solution without egg yolk (EW), the egg white solution with 0.1% egg yolk (EW–EY0.1), the egg white solution with 0.3% egg yolk (EW–EY0.3), the solution with 0.5% egg yolk (EW–EY0.5), and the egg white solution with 1% egg yolk (EW–EY1). After mixing the EW–EY solution of each group, transfer them to the stainless steel egg beater (DDQ-D01U1, GUANGDONG BEAR ELECTRIC CO., LTD) and beat them at 3 gear under normal temperature and pressure for 3 min. Each experiment was performed in triplicate.

The egg yolk concentration that began to significantly affect the foam characteristic was determined through preliminary experiments, and the EW–EY solution was treated by Lip-IM [22]. Lip-IM treatment with 0.1% (EWY 0.5% + Lip-IM-L), Lip-IM treatment with 0.5% (EWY 0.5% + Lip-IM-H), and the EW–EY solution without Lip-IM was used as the control (EWY 0.5%). After mixing each solution evenly, each experiment was performed in triplicate, according to the above methods. 

#### 2.4.2. Foaming Ability

The egg white foam samples were sent to the beakers and the scraps were scraped off the surface of the sample to keep the level and record the height of the sample. Each experiment was operated in triplicate. The foaming ability (*FA*) could be calculated by Equation (1):*FA* (%) = (*m*0 − *m*1)/*m*1 × 100(1)
where *m*0 is the mass of the sample solution that has not been stirred; *m*1 represents the mass of the sample solution of the same volume as *m*0 after being stirred.

#### 2.4.3. Foaming Stability

The foaming stability was measured by measuring the liquid outflow of the sample system. The samples were placed for 30 min, and the weight of the precipitated liquid was then assessed after it was carefully decanted. The more liquid precipitation, the worse the foam stability. The foaming stability (*FS*) could be analyzed by Equation (2):*FS* (%) = *mp*/*mi* × 100 (2)
where *mp* is the weight of the precipitated liquid; *mi* represents the initial quality of foam.

### 2.5. Molecular Dynamics Simulation

#### 2.5.1. Ab Initio Structure Modeling of Lip-IM

The amino acid sequence of Lip-IM was obtained from NCBI (FJ544454.1), and the signal peptide sequence of Lip-IM was predicted using the SignalP-5.0 [23] and resected manually. Then, the protein artificial intelligence prediction tool AlphaFold2 [24] was conducted to predict the structure of Lip-IM and the predicted structure pIDDTs scored 97.30 (Appendix A), indicating that the predicted structure is convincing.

#### 2.5.2. Long Time Molecular Dynamics Simulation under Different Temperature Conditions

The molecular dynamics simulation system of Lip-IM was performed at different temperatures (277, 293, 303, and 313 K) for 500 ns using the GROMACS 21.4 package [25] for a total of 4 molecular simulation systems with a total simulation time duration of 2.0 μs. Amber14sb_parmbsc1 force field and explicit solvation were used for each system [26]. Each system was enclosed within a cubic box filled with TIP3P water molecules. Counterions were introduced to achieve system charge neutrality, after which energy minimization by the steepest descent method was performed with a force constant of 1000.0 kJ/mol/nm. After that, conjugate gradient optimization (100.0 kJ/mol/nm) was used for the second energy minimization. Subsequently, the system was progressively equilibrated under canonical ensemble (NVT, 0.2 ns) and isothermal–isobaric ensemble (NPT, 0.2 ns) to ensure stability. Lastly, 500 ns production simulation was conducted for each MD system, with GPU acceleration using the RTX 3090. Except otherwise stated, other parameters were consistent with our previous methods [26].

#### 2.5.3. Molecular Dynamics Simulation Analysis

The root mean square deviation (RMSD), root mean square fluctuation analysis (RMSF), and solvent accessible surface area (SASA) of selected elements were calculated by gmx rms, gmx gyrate, and gmx sasa programs, respectively.

#### 2.5.4. Dynamic Cross-Correlation Matrices (DCCM)

In a protein, DCCM can be used to express motion correlations between certain amino acid atoms, for example, between Cα atoms and Cα atoms of other amino acids; these data provide information on the relevant kinematic properties of proteins at large scales [27]. All atoms’ dynamic motion descriptions are contained in the MD simulation trajectory, and dynamic correlation can be measured by computing the covariance between the two atomic fluctuations, as described by Equation (3): (3)c(i,j)=<ΔRi·ΔRj>
in which *i* and *j* are the sequence number of amino acid residues in the protein, *c*(*i*,*j*) is the covariance between *i* residues and *j* residues. is the atomic displacement vector, and the square brackets represent the global mean. Calculate the cross-correlation coefficient or normalized covariance using Equation (4):(4)$$c\left(i,j\right)=\frac{c\left(i,j\right)}{{\left[c\left(i,i\right)\cdot c\left(j,j\right)\right]}^{\frac{1}{2}}}$$

DCCM calculations fall between −1.0 and +1.0 (perfectly negative correlation and perfect correlation, respectively). The closer it gets to 1, the stronger the correlation; a plus or minus means the atoms are moving in the same direction. The closer *c*(*i*,*j*) is to 1, the stronger the correlation is, and a positive or negative value means the atoms are moving in the same direction.

#### 2.5.5. Shortest Path Map Analysis

The shortest path map (SPM) can study the effects of various temperatures on the interactions within proteins during the simulation, as well as the pathways by which key residues interact. The calculation of SPM was based on the research of Adrian et al. with slight modifications [28]. Briefly, the gromacs mdmat program was used to calculate the mean distance matrix between Cα atoms of Lip-IM, and 1.5 nm was used as the truncation distance between residues. Then, the mean distance matrix and DCCM matrix were imported into the igraph python (https://igraph.org/ accessed on 12 January 2023) package for calculation of SPM.

### 2.6. Statistical Analysis

Origin 2018 and SPSS ver. 11.5 were conducted for statistics analysis. Duncan’s multiple range test was performed to analyze statistical differences, and statistical significance was determined at a level of *p* < 0.05.

## 3. Results and Discussion

### 3.1. Expression and Purification of Lip-IM

The Lip-IM gene was successfully heterologously expressed in *E. coli* BL21 (DE3) with pET28a (+) as the vector. The crude enzyme solution was purified using Ni-NTA6FF and imidazole was removed through dialysis to obtain a pure enzyme solution. The molecular weight was estimated to be 22 KDa through SDS-PAGE analysis (Figure 1). The specific activity of Lip-IM was 11,048.18 U/mg. The results showed that the purity of the target protein was high enough to be used for subsequent enzymatic studies.

### 3.2. Characterization of Lip-IM

#### 3.2.1. The Effect of Temperature on the Activity and Stability of Lip-IM

To estimate the optimum temperature and stability of Lip-IM, the purified enzyme was exposed to various temperatures, and its activity was measured. The results revealed that the optimal temperature for Lip-IM was 30 °C, with more than 81.2% of the activity being maintained between 4–40 °C (Figure 2a). Additionally, the relative activity of Lip-IM was assessed after it was incubated at various temperatures for 3h to determine its stability. Lip-IM demonstrated the highest stability at 30 °C (Figure 2b), with a relative activity maintaining above 78.1% after 3h. Moreover, high relative activities were recorded at 4 °C and 20 °C, where the activity remained above 65%. Overall, Lip-IM exhibited high activity and stability at 4–30 °C. When the temperature increased to 50 °C, the stability began to decline, and the activity at 60 °C almost disappeared. Lip-IM may have lost its tertiary structure, which changed the configuration of active sites and reduced the interaction between Lip-IM and substrate, leading to decreased activity [29].

#### 3.2.2. The Effect of pH on the Activity and Stability of Lip-IM

The binding ability and catalytic activity of lipase to substrate molecules are affected by the pH, which can change the dissociation state of the substrate and lipase [30]. As shown in Figure 2c, Lip-IM has an optimal pH of 9, with more than 65% of its activity being retained within pH 8–11. However, the activity of Lip-IM was almost completely lost at a pH of 6. Additionally, Lip-IM showed the highest stability at pH 9. Any pH level below 8 or above 10 caused a significant decline in enzyme activity (Figure 2d). The activity of Lip-IM under acidic conditions was significantly lower than that under neutral and alkaline conditions. Therefore, Lip-IM can be considered a relatively stable alkaline lipase. It is noteworthy that the pH of an egg white system containing residual egg yolk is approximately 9, which is consistent with the optimum pH of Lip-IM. This strongly suggests that Lip-IM may have useful applications in the food industry.

#### 3.2.3. The Effect of Metal Ions on the Activity of Lip-IM

Metal ions can interact with the disulfide bond of an enzyme, altering its structure, or forming salts with the substrate to affect enzyme activity. Therefore, metal ions have a significant impact on enzyme activity [31]. As shown in Figure 3a, Na+, k+, Mg^2+^, and Cu^2+^ slightly inhibited the activity of Lip-IM at a concentration of 5 mM, but significantly inhibited it at a concentration of 10mM. The residual enzyme activity was about 62% when Zn^2+^ was present at 10 mM. On the other hand, Ca^2+^, Mn^2+^, Fe^2+^, Fe^3+^, and Cr^3+^ promoted the activity of Lip-IM to varying degrees. Among these, in comparison with the control group, Mn^2+^ promoted enzyme activity at a significant rate of 132%. The results indicate that metal ions can activate Lip-IM, by stabilizing its conformation and increasing its activity [32]. These findings are consistent with previous research. Maharana and Ray [33] demonstrated that Lip-IM activity is highest in the presence of Mn^2+^. Both Mn^2+^ and Ca^2+^ were found to be activators of lipase. Additionally, Ca^2+^ may remove fatty acids from the water–oil interface, increasing the solubility of oil and fat. As a result, lipase was able to bind oil molecules more effectively, thereby stimulating lipase-catalyzed oil hydrolysis [34].

#### 3.2.4. Substrate Specificity of Lip-IM

The results of substrate specificity of Lip-IM are shown in Figure 3b. Lip-IM exhibits high hydrolysis activity for pNP esters with short- and medium-chain fatty acids. Particularly, Lip-IM demonstrated the highest activity against pNP-caprylate(C8). While Lip-IM is capable of hydrolyzing pNP-derivatives with acyl chains as long as 16 and 18 carbons (pNP-palmitate, pNP-stearate), the observed activity level on substrates containing long-chain fatty acids was lower than that on substrates containing short- and medium-chain fatty acids. Compared with medium- and short-chain substrates, the enzyme activity of long-chain substrates remains above 50%. Due to its high specific activity, Lip-IM still exhibits high hydrolytic activity towards long-chain substrates.

#### 3.2.5. The Effect of Chemical Solvent on the Activity of Lip-IM

It can be seen from Table 1 that the activity of Lip-IM was not significantly affected by the organic solvents methanol, ethanol, isopropanol, acetonitrile, and DMSO (dimethyl sulfoxide), indicating that Lip-IM has good tolerance to organic solvents. No significant effect was also observed with the metal chelator (EDTA), with residual activity at approximately 90% after treatment with 10% EDTA. In contrast, SDS, Tween-80, and TritonX-100 considerably inhibited lipase activity. The inhibition caused by Tween-80 and Triton X-100 can be caused by the combination of surfactants with the secondary and tertiary structures of the enzyme, which caused the instability of protein conformation [35]. Among these, SDS exhibited the most substantial inhibitory effect on enzyme activity. After exposure to a 10% SDS solution, the residual activity dropped to only about 50%. This may be attributed to the anionic surfactant nature of SDS, which causes changes in the conformation of lipase and ultimately decreases its activity [36].

#### 3.2.6. Kinetic Parameters of Lip-IM

The study of enzyme kinetics provides valuable insight into the rate of enzyme-catalyzed reactions. In this regard, experiments were conducted using different concentrations of p-nitrophenol as a substrate, and the kinetic parameters of Lip-IM on C6, C8, C10, C12, C14, C16, and C18 were examined at 4–40 °C, and pH9.0, respectively. The measurement of *K*m reflects the affinity between enzyme and substrate. Overall, the results indicate that the *K*m of Lip-IM to short-chain substrates was the smallest across different temperature conditions, with the *K*m values of C6 and C8 being 3.92 ± 0.06 mM and 4.21 ± 0.09 mM at 30 °C, respectively. Moreover, the *K*m of C16 and C18 were the largest, suggesting that Lip-IM exhibited a higher affinity for short-chain substrates (C6, C8), followed by medium-long chain substrates (C10, C12, C14), whereas the affinity for long-chain substrates (C16, C18) was weaker, resulting in a larger substrate concentration required for the reaction. *k*cat is the catalytic constant of the enzyme, representing its ability to catalyze a specific substrate. Meanwhile, catalytic efficiency of an enzyme is measured by *k*cat/*K*m. As presented in Table 2, as carbon chain length increased, *k*cat/*K*m value decreased, indicating that lipase catalytic efficiency decreased.

### 3.3. Effect of Lip-IM on Foaming Properties of Egg White

#### 3.3.1. Foaming Properties

In food processing, egg yolk can significantly affect the foaming characteristics of egg white. Figure 4a illustrated the effect of yolk residue on the foaming characteristics. As the yolk residue increases, the foaming ability gradually decreases and the rate of liquid release exhibits an upward trend, which implies a gradual decrease in foam stability. Specifically, when the egg yolk residue was 0.5%, the foaming property experienced a significant decline, with the foaming ability decreasing from 487 to 362%, and the liquid precipitation rate increasing from 42.42 to 49.28%. These results indicate a notable decrease in foaming ability. Consequently, this residue was selected for the subsequent enzymatic hydrolysis experiment. The observed change was mainly attributed to the increased fat content in egg white solution as the amount of yolk residue increased.

#### 3.3.2. Enzymolysis Effect

To estimate the influence of lipase on the foaming characteristics and evaluate the potential application value of Lip-IM in the liquid egg industry, the foaming characteristics of egg white liquid (EW) without yolk, egg white liquid with yolk residue of 0.5% (EWY0.5%), sample system with a low proportion of lipase (EWY0.5% + Lip-IM-L), and sample system with a high proportion of lipase (EWY0.5% + Lip-IM-H) were measured, respectively.

As shown in Figure 4b, lipase can effectively enhance the foaming properties of the egg white system containing residual yolk to varying degrees. In contrast, heating alone without lipase cannot improve the whipping of the sample system under the same enzymatic hydrolysis conditions, indicating that enzymatic hydrolysis was not caused by heating alone.

The addition of 0.1% Lip-IM increased the foaming ability from 475 to 501%, while the 0.5% Lip-IM significantly improved the foaming ability by 85% (539%) compared to the pure egg white protein solution. Additionally, 0.5% Lip-IM also reduced the liquid precipitation rate of the sample system from 57 to 48%, indicating an improvement in foam stability. These results suggest that Lip-IM can effectively enhance the foaming characteristics of the egg white liquid system with residual egg yolk, and could serve as a promising biocatalyst in biotechnology applications.

### 3.4. Molecular Dynamics Simulation Analysis

#### 3.4.1. Basic Analysis

To investigate the stability and residue interaction of Lip-IM and study its structural changes, molecular dynamics (MD) simulations were conducted for 500 ns at 277, 293, 303, and 313 K. Specifically, MD simulations were conducted to elucidate the structural changes in Lip-IM at 4–40 °C, which facilitated the correlation between the observed lipase activity and the MD simulation results at corresponding temperatures [37].

A root mean square deviation analysis (RMSD) was performed to estimate the average deviation between the initial conformation of a protein and the final conformation, reflecting the stability of the system during the simulation process [26]. As shown in Figure 5a, the RMSD analysis of MD trajectory showed that Lip-IM had a small fluctuation in the simulation process of 277, 293, and 303 K, and the RMSD of 277 and 303 K systems reached a relative equilibrium after 100 ns, with the RMSD stable around 0.12 nm. However, during most of the MD simulation time at 313 K, the RMSD values were higher than other temperatures, indicating intense protein movement, and increased conformational flexibility, leading to decreased stability. These results suggest that Lip-IM at 40 °C could experience significant conformational changes that may impact Lip-IM function and activity. Wang, Wang, Xu, and Yu [38] reported that as the temperature increased, the flexibility of amino acids increased, leading to a decrease in enzyme activity. To improve thermostability, amino acids within the fluctuation region were mutated, and variant m31 processed RMSD values between 0 and 100 ns at lower rates than the wild type, indicating that the mutation increased the rigidity of the protein, improving its thermostability.

The analysis of the root mean square fluctuation (RMSF) involves calculating the average variation in atomic coordinates between different residues of a protein over the course of a simulation to determine the flexibility of specific regions within the protein. A higher RMSF value indicates greater flexibility of the amino acid at that site and a less stable region. The RMSF values of Lip-IM were analyzed at 277, 293, 303, and 313 K. As seen in Figure 5b, most regions showed stable vibration amplitudes with changing temperatures. However, at 313 K, there was a significant fluctuation in amino acids, predominantly around three local regions consisting of Gly93, Pro121, and Gly147 residue sites, as well as around Glu173–Asn183. To assess the relationship between these fluctuations and the active pocket, CavityPlus was utilized to scan the Lip-IM structure and predict its active pocket. These fluctuation regions were found to be in close proximity to the substrate entry site of the active pocket (Figure 5d). Wang, Wang, Xu, and Yu [38] demonstrated that increased flexibility of residues located in the active pocket could negatively affect substrate binding stability, thereby reducing enzyme catalyzed reaction rates. Thus, we hypothesized that catalytic activity of Lip-IM decreased at high temperatures, as a result of the increased flexibility of pocket residues. When temperatures were lowered, the flexibility of the Lip-IM region was reduced, which helped stabilize substrate binding and catalysis.

As a measure of protein hydrophobicity, the solvent-accessible surface area (SASA) can be used to determine the surface area of biomolecules that are accessible to solvent molecules [39]. Figure 5c shows that SASA fluctuated between 79.92–93.17 nm^2^ at 277 K, 80.58–94.73 nm^2^ at 293 K, 79.60–93.26 nm^2^ at 303 K, and 80.97–96.45 nm^2^ at 313 K. As the temperature increased, SASA values fluctuated significantly, particularly at 313 K, indicating that the surface area of Lip-IM exposed at higher temperatures significantly increased. Increased amino acid residue flexibility due to increased temperature promoted interaction between amino acid residues and water, facilitating hydration. Conversely, at lower temperatures, hydrophobic interactions can help maintain protein spatial conformation.

#### 3.4.2. Lip-IM Dynamic Cross-Correlation Map (DCCM)

To further investigate the impact of temperature on amino acid movement correlation in Lip-IM and to identify regions dynamically correlated with flexible active sites, dynamic cross-correlation maps (DCCM) were generated for Lip-IM at various temperatures. Regions in red indicate a high positive correlation while regions in blue indicate a high negative correlation.

Analysis of the DCCMs revealed that the correlation maps of amino acids at binding sites at 277 K (Figure 6a) and 293 K (Figure 6b) were similar to those seen at 303 K (Figure 6c). In contrast, the DCCM generated for 313 K (Figure 6d) showed increased overall correlation between the binding sites (namely the 93rd, 121st, 147th, and 173rd–183rd residues) and other amino acids. There was a strong positive correlation between the region around Gly93 and the region around Pro121, while Pro121 exhibited a strong positive correlation with the region around Gly147. This contributed towards the stabilization of the regions Val73–Asn183 at 313 K.

#### 3.4.3. Lip-IM Shortest Path Map

The shortest path map (SPM) is a useful tool for analyzing mutual force networks within a protein structure. The density and number of points and bonds in the SPM reveal the overall stability of the structure, with higher values indicating stronger stability. Figure 7 shows that the internal interaction network of the protein was tightly packed at 277 K, 293 K, and 303 K, indicating that the interaction force was relatively strong, and the protein structure was stable. However, at 313 K, the interaction force was weakened, leading to a sparse internal interaction network. This was accompanied by significant movement in the positions of Glu173 and Gly174, which may have been caused by the unfolding of the protein structure due to high temperature. This observation was also consistent with the changes seen in SASA.

## 4. Conclusions

In this study, Lip-IM demonstrated high hydrolysis activity for long chain fatty acids (C16, C18) and exhibited the highest enzyme activity at 30 °C and pH 9. Importantly, the lipase activity was also maintained at around 80% when stored between 4–20 °C. When used to treat egg white liquid that had been separated from the yolk, a significant improvement in foaming ability and stability was achieved by adding lipase, with the restoration of more than 80% of the original foam properties. This suggests that Lip-IM can enzymatically hydrolyze the lipid components in the remaining yolk, thus improving the performance of the egg white liquid. Furthermore, molecular dynamics simulations were conducted on Lip-IM to explore its structural stability and residue interactions. Results from the MD simulations showed that regions comprising Gly93, Pro121, Gly147, and Glu173–Asn183 in Lip-IM were sensitive to high temperature, exhibiting significant structural flexibility. Notably, the flexible regions unfolded locally during the simulation at 313 K, which could be detrimental to substrate binding. Collectively, this study provides important insights into the temperature–structural-stability relationship of Lip-IM and identifies a promising lipase for use in the food industry. These findings have paved the way for Lip-IM modification and practical applications.

## Figures and Tables

**Figure 1 foods-12-04098-f001:**
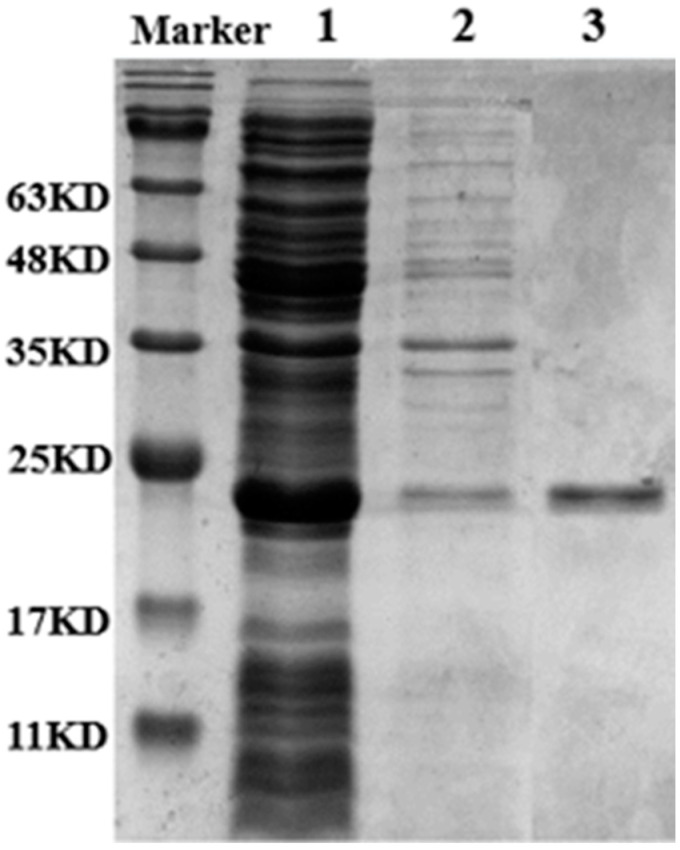
SDS-PAGE analysis of the purities of Lip-IM. Lane 1: supernatant after Lip-IM fragmentation after induced expression. Lane 2: precipitation after Lip-IM fragmentation after induced expression. Lane 3: purified Lip-IM.

**Figure 2 foods-12-04098-f002:**
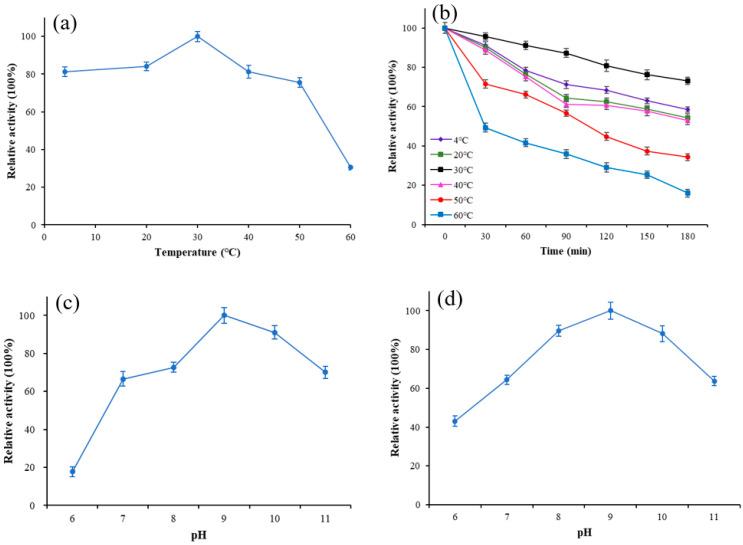
The effect of temperature and pH on activity of Lip-IM. (**a**) The optimization of temperature. (**b**) Thermal stability. (**c**) The optimization of pH. (**d**) pH tolerance.

**Figure 3 foods-12-04098-f003:**
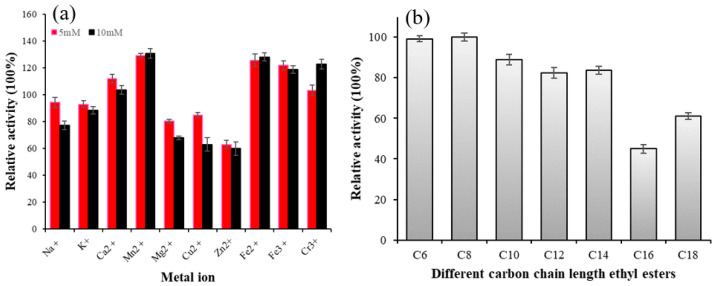
(**a**) The effects of different concentrations of metal ions on lipase activity. (**b**) Substrate specificity of Lip-IM to pNP esters of varying C chain lengths.

**Figure 4 foods-12-04098-f004:**
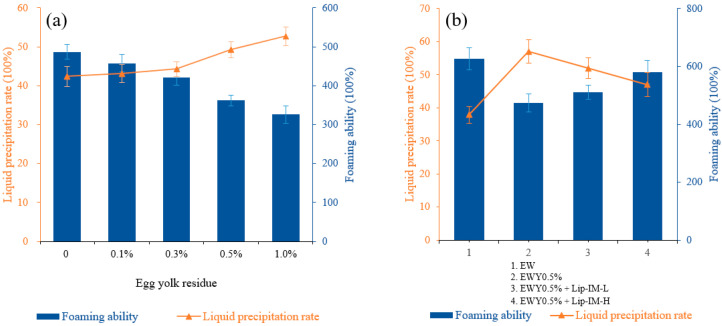
(**a**) The effect of different concentrations of egg yolk residue on foaming ability and foaming stability of egg white liquid. (**b**) The effect of different concentrations of Lip-IM on foaming ability and foaming stability of egg white liquid.

**Figure 5 foods-12-04098-f005:**
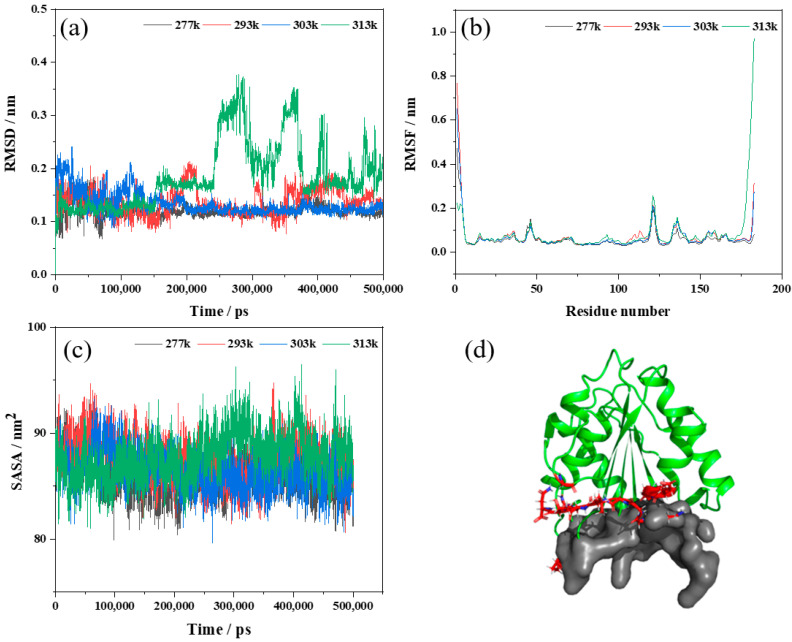
Molecular dynamics simulation results of Lip-IM at different temperatures (277, 293, 303, and 313 K). (**a**–**c**) represent the root mean square deviation (RMSD), the root mean square fluctuation (RMSF) and solvent-accessible surface area (SASA), respectively; (**d**) represents the active pocket prediction of Lip-IM.

**Figure 6 foods-12-04098-f006:**
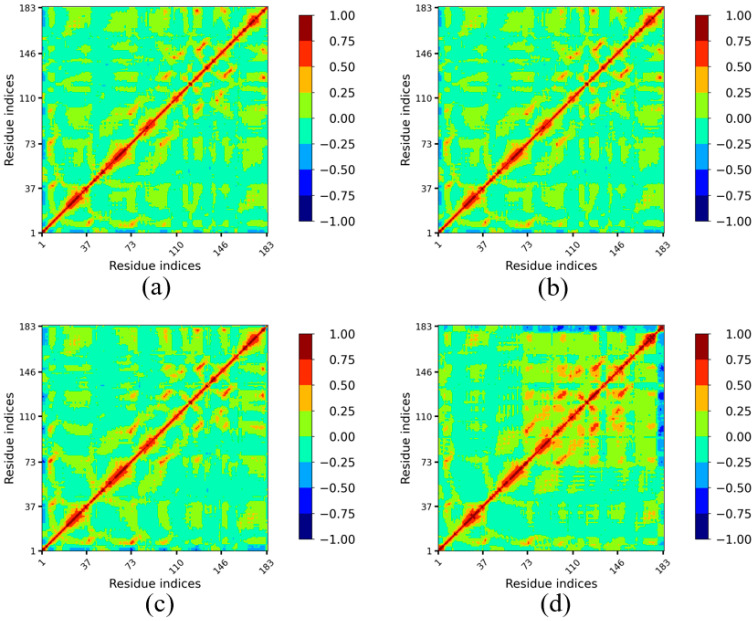
Dynamic cross-correlation plot of Lip-IM at different temperatures. The color scheme runs between blue (−), green (0) and red (+). A negative value indicates anti-correlation i.e., the atom displacement is in the opposite direction, and a positive value indicates correlated motion i.e., atom displacement in the same direction.

**Figure 7 foods-12-04098-f007:**
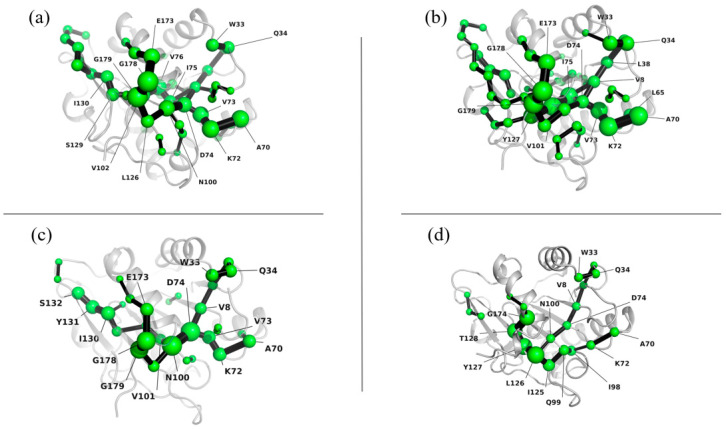
Shortest path map of Lip-IM at 277 K (**a**), 293 K (**b**), 303 K (**c**), and 313 K (**d**).

**Table 1 foods-12-04098-t001:** Effect of different concentrations of chemical solvents on the enzymatic activity of Lip-IM.

Chemical Solvent	Relative Activity (%)	
	5%	10%
Methanol	92.51 ± 2.17 a	86.30 ± 2.09 a
Ethanol	97.83 ± 1.79 a	93.74 ± 1.59 a
Isopropanol	93.24 ± 2.04 a	87.67±2.18 a
Acetonitrile	98.16 ± 1.87 a	96.08 ± 1.58 a
DMSO	91.76 ± 0.98 a	89.32 ± 1.80 a
EDTA	92.16 ± 1.37 a	87.63 ± 2.01 a
SDS	62.38 ± 0.79 b	49.29 ± 1.27 c
Tween-80	76.15 ± 1.56 b	61.07 ± 2.15 b
Triton X-100	87.49 ± 2.14 ab	73.36 ± 1.69 b

Different lowercase letters represent significant differences at *p* < 0.05 determined using one-way ANOVA with Duncan’s multiple range test.

**Table 2 foods-12-04098-t002:** Kinetic parameters of Lip-IM at different temperatures.

Temperature	Carbon Chain Length	*V*max (μM min^−1^)	*K*m (mM)	*k*cat (S^−1^)	*k*cat/*K*m (S^−1^ μM^−1^)
4 °C	C6	211.52 ± 3.68	4.41 ± 0.06	1.94 ± 0.05	439.23 ± 3.79
	C8	210.43 ± 2.79	4.62 ± 0.09	1.93 ± 0.07	417.75 ± 4.57
	C10	187.25 ± 2.94	5.19 ± 0.04	1.71 ± 0.06	329.48 ± 3.27
	C12	246.37 ± 3.18	5.32 ± 0.08	2.26 ± 0.06	424.81 ± 4.13
	C14	175.34 ± 2.57	5.63 ± 0.07	1.61 ± 0.04	285.20 ± 2.59
	C16	194.32 ± 2.33	7.62 ± 0.06	0.79 ± 0.01	103.67 ± 1.78
	C18	230.64 ± 2.61	7.21 ± 0.05	0.94 ± 0.02	130.37 ± 1.96
20 °C	C6	217.79 ± 2.93	4.16 ± 0.06	1.99 ± 0.05	478.37 ± 4.17
	C8	223.20 ± 2.06	4.43 ± 0.09	2.04 ± 0.07	460.50 ± 4.36
	C10	196.41 ± 2.31	5.04 ± 0.04	1.80 ± 0.06	357.14 ± 3.95
	C12	214.43 ± 2.64	5.77 ± 0.08	1.96 ± 0.06	339.69 ± 3.26
	C14	189.43 ± 1.98	5.40 ± 0.07	1.74 ± 0.05	322.22 ± 3.42
	C16	157.79 ± 1.67	8.74 ± 0.08	0.64 ± 0.01	73.33 ± 1.06
	C18	278.40 ± 2.14	6.95 ± 0.07	1.13 ± 0.02	162.59 ± 2.02
30 °C	C6	232.69 ± 2.47	3.92 ± 0.06	2.13 ± 0.05	543.59 ± 4.95
	C8	236.25 ± 2.29	4.21 ± 0.09	2.16 ± 0.07	513.89 ± 4.16
	C10	184.21 ± 1.94	5.23 ± 0.04	1.69 ± 0.06	322.54 ± 3.97
	C12	204.47 ± 2.18	5.96 ± 0.08	1.87 ± 0.06	314.17 ± 3.65
	C14	213.62 ± 2.38	5.12 ± 0.09	1.96 ± 0.05	382.08 ± 3.97
	C16	150.26 ± 2.02	9.26 ± 0.09	0.61 ± 0.01	65.87 ± 1.17
	C18	292.43 ± 3.12	6.26 ± 0.07	1.19 ± 0.04	190.10 ± 2.18
40 °C	C6	221.64 ± 3.50	4.27 ± 0.06	2.03 ± 0.04	475.33 ± 4.93
	C8	243.75 ± 2.64	3.98 ± 0.05	2.23 ± 0.06	560.84 ± 5.71
	C10	178.69 ± 2.33	5.43 ± 0.04	1.64 ± 0.01	301.35 ± 4.05
	C12	186.79 ± 2.76	6.78 ± 0.08	1.71 ± 0.02	252.30 ± 3.28
	C14	170.16 ± 2.62	5.59 ± 0.05	1.56 ± 0.03	278.76 ± 3.37
	C16	137.42 ± 2.25	11.07 ± 0.09	0.56 ± 0.01	50.59 ± 1.94
	C18	247.68 ± 3.47	7.14 ± 0.07	1.00 ± 0.04	140.06 ± 2.56

## Data Availability

The data presented in this study are available on request from the corresponding author.

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
