# Peer review of "Characterization and Molecular Dynamics Simulation of a Lipase Capable of Improving the Functional Characteristics of an Egg-Yolk-Contaminated Liquid Egg White"

_foods, 2023, doi:10.3390/foods12224098_

Round 1

Reviewer 1 Report

Comments and Suggestions for Authors

Authors describes the characterizations of Lip-IM for hydrolyzing shorter chain lipids with higher specificity. The manuscript is well-written with detailed introduction and discussion. I have two minor comments:

1. Figure 4b. 0.5% Lip-IM shows to improve the foaming ability of the EWY0.5% group by 85%. But it's still lower than the EW group. Did authors tried adding more Lip-IM in EWY0.5% to restore the foaming ability as the pure EW?

2. Foaming ability features of egg white treated with Lip-IM has been evaluated well. But after Lip-IM treatment, how about other quality characteristics of egg white? For instance. is the treated egg white still as nutritious as untreated? Is the shelf-life shorter after treatment? 

Author Response

Dear Reviewer,

Thanks for your great concern and comments on my initial manuscript. Complying with your comments, the answers to each comment have been addressed as follows.

Comments 1: Figure 4b. 0.5% Lip-IM shows to improve the foaming ability of the EWY0.5% group by 85%. But it's still lower than the EW group. Did authors tried adding more Lip-IM in EWY0.5% to restore the foaming ability as the pure EW?

Response 1: In our preliminary experiment, we observed that when the addition of Lip-IM reaches 5%, the foaming ability of EWY can be restored to approximately 90%. Considering production costs, we chose a lower addition amount (0.5%) that could significantly improve the foaming ability. Therefore, it is particularly important to modify Lip-IM to enhance its catalytic efficiency, which is also the focus of our future research. Thank you very much.

Comments 2: Foaming ability features of egg white treated with Lip-IM has been evaluated well. But after Lip-IM treatment, how about other quality characteristics of egg white? For instance. is the treated egg white still as nutritious as untreated? Is the shelf-life shorter after treatment?

Response 2: Thank you very much for your suggestion. The problem mentioned by the reviewer is a matter of concern. In the latter study we will study the other quality characteristics of the treated egg white. Thank you again.

Reviewer 2 Report

Comments and Suggestions for Authors

  The authors of this manuscript (Manuscript ID foods-2684187) report interesting and novel data. The utilized, by the authors, main methodologies are the appropriate ones and they produced a big volume of substancial work (gene expression, protein purification, effect of lipase on foaming properties, Molecular dynamics simulation (comprising ab initio structure modeling of lipase, long time molecular dynamics simulation, etc, .. etc)

  However, I am skeptic. Why authors try to conclude about several properties of their lipase (enzymatic assays, effects of temperature, pH-value, metallic ions, organic solvents, etc .. etc, on the properties of lipase) by using old fashion and unreliable methodologies?

  Figures 2 and 3, as well as table 1, do not offer value to the manuscript under consideration. On the contrary, they show a kind of carelessness and irresponsibility for the authors; I am sure that this is not the case, and that authors just did some "boring" extra experiments.

  Therefore, authors should proceed accordingly and improve the experimental part in order to earn scientifically approved results (subsections: 2.3.1. - 2.3.7). The term "Relative Activity (100%) does mean nothing. Instaed, entities like Vmax (or kcat), Km, and their ratio, are the ONLY acceptable; moreover that authors report results concerning the Vmax, and Km values of their lipase. Another important mistake of authors is the presentation of their results on the temperature and pH-value profiles; acctually, these profiles offer only a small percentage of the scientific information contained within the experiental points.

   Nevertheless, authors should revise their manuscript by:

1) Repeat their experiments (I refer to subsections: 2.3.1. - 2.3.7), under conditions comprising either [S] << Km, or [S] >> Km (or better both of them).

2) Prepare new temperture and pH-value profiles whose ordinates should be either the Vmax or the Km values (NOT the "relative activity"), and/or their ratio. The abscissae of the temperature profile should be the absolute temperature. In both cases authors should use the proper equations, in order to fit their experimental data. These nonlinear regressions will support the manuscript with very important thermodunamic and other valuable information.

Authors should also correct the words: "mental: or metal to metallic and "iron" to ions (subsection: 2.3.4.).

Overall: I suggest a  revision of this manuscript, according to my comments, and reconsideration of its revised version.

Comments on the Quality of English Language

moderate editing of English language required.

Author Response

Dear Reviewer,

  Thanks for your great concern and comments on my initial manuscript. Complying with your comments, we have revised the manuscript carefully. The answers to each comment have been addressed as follows.

Comments 1: Repeat their experiments (I refer to subsections: 2.3.1. - 2.3.7), under conditions comprising either [S] << Km, or [S] >> Km (or better both of them).

Response 1: Thank you for this suggestion. Before conducting the experiment, we had extensively reviewed the literature, and the experimental method we employed was widely used by many researchers.

References

Zhao, J., Ma, M., Yan, X., Zhang, G., Xia, J., Zeng G., et al. Expression and characterization of a novel lipase from Bacillus licheniformis NCU CS-5 for application in enhancing fatty acids flavor release for low-fat cheeses. Food Chem. 2022, 30, 130868. https://doi.org/10.1016/j.foodchem.2021.130868.

Xing, S., Zhu, R., Li, C., He, L., Zeng, X. Zhang, Q. Gene cloning, expression, purification and characterization of a sn-1,3 extracellular lipase from Aspergillus niger GZUF36. J Food Sci. 2020, 57, 2669-2680. https://doi.org/10.1007/s13197-020-04303-x

Xiang, M., Wang, L., Yan, Q., Jiang, Z., Yang, S. Heterologous expression and biochemical characterization of a cold-active lipase from Rhizopus microspores suitable for oleate synthesis and bread making. Biotechnol. Lett. 2021, 43, 1921-1932. https://doi.org/10.1007/s10529-021-03167-1

Zhang, Y., Ji, F., Wang, J., Pu, Z., Jiang, B., Bao, Y. Purification and characterization of a novel organic solvent‑tolerant and cold‑adapted lipase from Psychrobacter sp. ZY124. Extremophiles. 2018, 22, 287-300. https://doi.org/10.1007/s00792-018-0997-8

Gokbulut, A., Arslanoglu, A. Purification and biochemical characterization of an extracellular lipase from psychrotolerant Pseudomonas fluorescens KE38.   Turk. J. Biol. 2013, 37, 538-546. https://doi:10.3906/biy-1211-10

Jiang, C. Alteration of substrate specificity of lipase and its application in hydrolysis of oils and fats. Master's thesis, Jiangnan University, Wuxi, 2018.

Comments 2:  Prepare new temperature and pH-value profiles whose ordinates should be either the Vmax or the Km values (NOT the "relative activity"), and/or their ratio. The abscissae of the temperature profile should be the absolute temperature. In both cases authors should use the proper equations, in order to fit their experimental data. These nonlinear regressions will support the manuscript with very important thermodynamic and other valuable information.

Response 2:  Thank you for this suggestion. In previous research, the abscissa of the temperature profile was “temperature”, and the ordinates of temperature and pH-value profiles were “relative activity”. Therefore, we think the coordinates of our graph are reasonable. At this point, we do not have the necessary to prepare new temperature and pH-value profiles. Thank you again.

References

Zhang, X-F., Ai, Y-H., Xu, Y., Yu, X-W. High-level expression of Aspergillus niger lipase in Pichia pastoris: Characterization and gastric digestion in vitro. Food Chem. 2019, 274, 305-313.

Zhao, J., Ma, M., Yan, X., Zhang, G., Xia, J., Zeng G., et al. Expression and characterization of a novel lipase from Bacillus licheniformis NCU CS-5 for application in enhancing fatty acids flavor release for low-fat cheeses. Food Chem. 2022, 30, 130868. https://doi.org/10.1016/j.foodchem.2021.130868.

Cheng, Y-Y., Qian, Y-K., Li, Z-F., Wu, Z-H., Liu, H., Li, Y-Z. A Novel Cold-Adapted Lipase from Sorangium cellulosum Strain So0157-2: Gene Cloning, Expression, and Enzymatic Characterization. Int. J. Mol. Sci. 2011, 12(10), 6765-6780. https://doi.org/10.3390/ijms12106765

Comments 3: Authors should also correct the words: "mental: or metal to metallic and "iron" to ions (subsection: 2.3.4.).

Response 3: Thank you very much for your reminder. We have corrected “mental irons” to “metal ions”. Thank you again.

Reviewer 3 Report

Comments and Suggestions for Authors

The manuscript entitled "Characterization and molecular dynamics simulation of a lipase capable of improving the functional characteristics of an egg yolk-contaminated liquid egg white" written by Xu et al., presents the characterization and the performances of Lipase Baccilus subtilis

The data are clear and in a logic form presented.

Before publication I would suggest some minor modifications:

Line 16 Bacillus subtilis must be written Italic Bacillus subtilis.

Lines 17, 107,119, 280, E. coli must be italic E. coli. Please check the entire document. 

Line 145 the OD  from OD410 must be removed. Generally OD is used for optical density. 

Thesubchapters titles 3.2.4 Substrate specificity of Lip-IM (line 335) and 3.2.6 Substrate specificity of Lip-IM are similar. 

Author Response

Dear Reviewer,

  Thanks for your great concern and comments on my initial manuscript. Complying with your comments, we have revised the manuscript carefully. The answers to each comment have been addressed as follows.

Comments 1: Line 16 Bacillus subtilis must be written Italic Bacillus subtilis.

Response 1: Line 16: Thank you very much for your reminder. We have modified it to Italic “Bacillus subtilis”.

Comments 2: Lines 17, 107,119, 280, E. coli must be italic E. coli. Please check the entire document. 

Response 2: Lines 17, 87, 107, 119, 279: Thank you very much for your reminder. We have checked the entire document and revised “E. coli” to “E. coli.”. Thank you again.

Comments 3: Line 145 the OD from OD410 must be removed. Generally OD is used for optical density. 

Response 3: Line 145: We have deleted “OD”. Thank you very much.

Comments 4: The subchapters titles 3.2.4 Substrate specificity of Lip-IM (line 335) and 3.2.6 Substrate specificity of Lip-IM are similar. 

Response 4: Line 363: Thank you very much for your reminder. We have changed “3.2.6 Substrate specificity of Lip-IM” to “3.2.6 Kinetic Parameters of Lip-IM”. Please find this revision in the manuscript. Thank you again.